# Exercise Cardiac Load and Autonomic Nervous System Recovery during In-Season Training: The Impact on Speed Deterioration in American Football Athletes

**DOI:** 10.3390/jfmk8030134

**Published:** 2023-09-12

**Authors:** Eric Renaghan, Harrison L. Wittels, Luis A. Feigenbaum, Michael Joseph Wishon, Stephanie Chong, Eva Danielle Wittels, Stephanie Hendricks, Dustin Hecocks, Kyle Bellamy, Joe Girardi, Stephen Lee, Tri Vo, Samantha M. McDonald, S. Howard Wittels

**Affiliations:** 1Department of Athletics, Sports Science, University of Miami, Miami, FL 33146, USA; eric.renaghan@miami.edu (E.R.); lfeigenbaum@med.miami.edu (L.A.F.); 2Tiger Tech Solutions, Inc., Miami, FL 33156, USA; hl@tigertech.solutions (H.L.W.); joe@tigertech.solutions (M.J.W.); schong591@gmail.com (S.C.); evadanielle@gmail.com (E.D.W.); steph.hendricks@gmail.com (S.H.); dustin@tigertech.solutions (D.H.); shwittels@gmail.com (S.H.W.); 3Department of Physical Therapy, Miller School of Medicine, University of Miami, Miami, FL 33146, USA; j.girardi@miami.edu; 4Department of Athletics, Nutrition, University of Miami, Miami, FL 33146, USA; k.bellamy1@umiami.edu; 5United States Army Research Laboratory, Adelphi, MD 20783, USA; stephen.j.lee28.civ@mail.mil; 6Navy Medical Center-San Diego, San Diego, CA 92134, USA; huuv@g.clemson.edu; 7School of Kinesiology and Recreation, Illinois State University, Normal, IL 61761, USA; 8Department of Anesthesiology, Mount Sinai Medical Center, Miami, FL 33140, USA; 9Department of Anesthesiology, Wertheim School of Medicine, Florida International University, Miami, FL 33199, USA; 10Miami Beach Anesthesiology Associates, Miami, FL 33140, USA

**Keywords:** exercise training, overtraining, sports, strength and conditioning, collegiate

## Abstract

Fully restoring autonomic nervous system (ANS) function is paramount for peak sports performance. Training programs failing to provide sufficient recovery, especially during the in-season, may negatively affect performance. This study aimed to evaluate the influence of the physiological workload of collegiate football training on ANS recovery and function during the in-season. Football athletes recruited from a D1 college in the southeastern US were prospectively followed during their 13-week “in-season”. Athletes wore armband monitors equipped with ECG and inertial movement capabilities that measured exercise cardiac load (ECL; total heartbeats) and maximum running speed during and baseline heart rate (HR), HR variability (HRV) 24 h post-training. These metrics represented physiological load (ECL = HR·Duration), ANS function, and recovery, respectively. Linear regression models evaluated the associations between ECL, baseline HR, HRV, and maximum running speed. Athletes (n = 30) were 20.2 ± 1.5 years, mostly non-Hispanic Black (80.0%). Negative associations were observed between acute and cumulative exposures of ECLs and running speed (β = −0.11 ± 0.00, *p* < 0.0000 and β = −0.15 ± 0.04, *p* < 0.0000, respectively). Similarly, negative associations were found between baseline HR and running speed (β = −0.45 ± 0.12, 95% CI: −0.70, −0.19; *p* = 0.001). HRV metrics were positively associated with running speed: (SDNN: β = 0.32 ± 0.09, *p* < 0.03 and rMSSD: β = 0.35 ± 0.11, *p* < 0.02). Our study demonstrated that exposure to high ECLs, both acutely and cumulatively, may negatively influence maximum running speed, which may manifest in a deteriorating ANS. Further research should continue identifying optimal training: recovery ratios during off-, pre-, and in-season phases.

## 1. Introduction

Optimal sports performance requires complete recovery of the autonomic nervous system (ANS) [1]. The ANS regulates many physiological processes involved in athletic performance such as skeletal muscle contraction, cardiac function, and vascular compliance [2]. As such, the functionality of the ANS affects performance metrics such as speed, agility, reaction time, force production, and power output [3,4]. For sports such as collegiate football, the repeated powerful movements profoundly impact the ANS, often leading to prolonged sympathetic nervous system dominance [5,6]. Full recovery of the ANS is paramount for peak sports performance during competitions [5]. Failing to provide adequate recovery, especially during in-season, invites negative consequences such as non-functional overtraining, and decrements in sports performance, all signs of a deteriorating ANS [7,8].

The window of recovery following competitions varies for each sport and depends on its duration, intensity, etc. For contact sports such as collegiate football, athletes endure, for nearly four hours, heightened levels of adrenaline, maximal force production, and power output [5]. These prolonged competitions result in augmented physical and mental fatigue, skeletal muscle damage, energy depletion, and muscular soreness. As such, football athletes may require at least 72 h for full recovery [5,9,10,11,12,13]. Sport performance researchers recommend markedly reducing training volume and limiting high-intensity training during the in-season allowing for sufficient post-competition recovery [14]. While strong evidence supports the necessity of recovery, reports suggest that many coaches fear reducing training volume as it may detrain athletes, resulting in poor performance [6]. Interestingly, studies show detraining occurs during the in-season however, most significantly among “reserve” or “bench” players, the athletes who are minimally exposed to competitive play [6]. The trepidation of detraining likely leads to coaches training athletes at higher volumes and intensities. Consequently, studies also show that higher training loads during the in-season predispose athletes to injury (e.g., ligament tears or muscle strains), likely a result of a deteriorating ANS [15]. 

A significant limitation of the former research, however, is the limited number of studies on collegiate football, a sport played by nearly 25% of all NCAA athletes [16]. Studies examining the influence of training volume during the in-season in contact sports primarily focused on soccer and rugby athletes [17,18]. Comparatively, collegiate football differs considerably as competitions are 50 to 60% longer and pose a greater risk for severe injuries. This risk is likely exacerbated if training regimens do not account for supramaximal efforts performed during football competitions, however, this remains unclear [6]. Therefore, this study aimed to evaluate the influence of the physiological demand imposed upon collegiate football athletes during the in-season and the recovery and function of the ANS. Specifically, we examined the association between the exercise cardiac load (total heartbeats) endured during the preceding week’s training sessions and ANS recovery and function among collegiate football athletes. We hypothesized that the sustained, hyperbolic cardiac load endured among competing athletes would compound the impact of the weekly training sessions on the ANS throughout the in-season. Specifically, we anticipated observing a negative association between 24-h baseline heart rate (HR) and maximum running speed and a positive association between heart rate variability (HRV) and maximum speed, each representing ANS recovery and function, respectively. 

## 2. Materials and Methods

### 2.1. Study Design

The current study employed a 13-week, prospective study design among a sample of Division I collegiate male football athletes during their “in-season” training. The physiological load of weekly training was estimated using an exercise cardiac load metric. ANS recovery was measured using baseline HR and HRV and ANS function was estimated via the athletes’ top speed reached during weekly trainings. 

### 2.2. Subjects

Subjects were recruited from a Division I collegiate football team located in the southeastern region of the State of Florida. The athletes were participating in their routine 13-week “in-season” training program. Practice sessions included aerobic, speed, strength, agility, and power-focused exercises. While each training session varied daily and weekly, the athletes consistently engaged in moderate-vigorous intensity exercise lasting between 120 to 180 min every, Tuesday, Wednesday, Thursday, Friday, and Sunday (Figure 1). The prospective participants were recruited from a pre-selected group of athletes the coaches identified as “starters”, which were athletes that competed in nearly every regulation game and for most of its duration. Starters were recruited as they endured a greater physiological load during a given week consequent to participating in weekly competitions. Additionally, due to the wide variability in movement patterns across player positions, only starting athletes playing “heavy running” positions including cornerback, running back, tight end, and wide receiver were included in the primary analyses. Prior to any measurements, the athletes were informed of the benefits and risks of the study and the conflicts of interest of all authors. All athletes participating voluntarily consented to the study. All study protocols followed the ethical principles defined in the Declaration of Helsinki and were approved by the university’s Institutional Review Board (IRB #20191223). 

### 2.3. Methodology

#### 2.3.1. Cardiac and ANS Measurement

Thirty participants were fitted with armband monitors equipped with temperature, electrocardiography (ECG), photoplethysmography (PPG), and inertial measurement unit (IMU) capabilities (Warfighter Monitor (WFM), Tiger Tech Solutions Inc., Miami, FL, USA). The WFM armbands were previously validated in several diverse subpopulations [19]. Monitors were placed on the posterior aspect of the left upper arm, secured with an elastic band, and worn at the start and throughout each training session (n = 128). Although the WFM device collected several biometric parameters, only cardiac function and IMU data were analyzed.

#### 2.3.2. Exercise Cardiac Load during In-Season Weekly Training 

Exercise cardiac load (ECL) represented the physiological load athletes endured during each training session. ECL was the product of the athlete’s average HR (bpm) and duration (minutes) of weekly training sessions. Both HR and duration are strong contributors to physiological load during exercise [14]. Only HRs sustained at ≥85 bpm were calculated for average HR as this threshold was considered “active training”. ECL was normalized with the largest ECL measured, from any athlete, during the in-season and multiplied by 100 for purposes of correlation.
$$ECL\;\left(total\;heartbeats\right)=Average\;HR(bpm)\times Session\;Duration\;(minutes)$$

#### 2.3.3. ANS Recovery

Next-day baseline HR represented ANS recovery. Baseline HR was measured in the early morning and following at least 4 min of inactivity, per established protocols [20]. Specifically, baseline HR was measured prior to the start (0600–0700) of the following day’s exercise training session. Each athlete was required to remain nearly motionless in a seated position for a period of 5 min to collect a “resting” baseline HR.

The HRV metrics used included the standard deviation of NN intervals (SDNN) and the root mean square of successive differences (rMSSD). HRV was measured during the same 5-min interval as baseline HR. Further details on these metrics are described elsewhere [21].

#### 2.3.4. Maximum Running Speed

Maximum running speed served as the outcome variable and was defined as the fastest recorded speed in miles per hour (mph) by an athlete during a single training session. Speed was calculated using a nine-degree-of-freedom inertial measurement unit (9-DOF IMU) on the WFM. The 9-DOF IMU provides a three-axis accelerometer, gyroscope, and magnetometer. Utilizing the magnetometer and the accelerometer the normal vector (*z*-axis) was identified and gravity was removed to give us the remaining accelerometer data which contains the two-dimensional, x- and y-plane of accelerations. Utilizing the gyroscope, the x and y accelerometer values were forced to zero during non-movement periods. Further, with the gyroscope, the dominant movement direction within the x and y planes was identified. We then, integrated the accelerometer data with a starting value of zero along the dominant direction in the x-y plane to quantify velocity. To calculate absolute speed, the directional component was removed [22,23]. 

### 2.4. Statistical Analyses

The current study evaluated the associations between ECL, HRV, baseline HR of both acute and cumulative exposures to in-season training, and its influence on maximum running speed. For acute training, (e.g., daily sessions) ECLs, HRV, and baseline HR values were averaged across daily sessions for each of the 13 weeks of in-season training. For the cumulative exposures, ECL was averaged over one training week. Maximum running speed served as the primary outcome variable. Associations were quantified using linear regression models and were performed separately for each metric. The normality of the conditional distributions was assessed via the Kolmogorov-Smirnov test and was deemed normally distributed. For all models, β coefficients and standard errors were estimated, and the *a priori* threshold for statistical significance was set at α = 0.05. Statistical analyses were performed in MATLAB, version 2021b (MathWorks, Natick, MA, USA). 

## 3. Results

The descriptive characteristics of the sample of D1 collegiate football athletes are presented in Table 1. Of all the starters (n = 30), 16.7%, 23.3%, 13.3%, and 20.0% were cornerbacks, running backs, tight ends, and wide receivers, respectively. Athletes were, on average, 20.2 ± 1.5 years of age, predominantly non-Hispanic Black 80.0%, and had an average body mass index of 27.6 ± 2.3 kg/m^2^ and ranging from 23.7 to 32.5 kg/m^2^. 

Table 2 shows the weekly average and 50th percentiles for the average exercise cardiac load of in-season training sessions and the recovery and function of the athletes’ ANS. The acute and cumulative ECL of in-season training were 21,800.0 ± 4600.0 and 108,700.0 ± 22,800.0 total heartbeats, respectively. On average, the next-day baseline HR was 60.9 ± 8.6 bpm and ranged from 48.8 to 112.2 bpm. The average maximum running speed achieved across 25 weeks of in-season training was 17.3 ± 1.4 mph and ranged from 15.0 to 22.0 mph.

The correlation coefficients for the associations between exercise cardiac load, ANS recovery, and maximum running speed during in-season training are presented in Table 3. Statistically significant, negative associations between both acute and cumulative exposure to exercise cardiac loads and maximum running speed achieved during in-season training (acute: β = −0.11 ± 0.00, 95% CI: −0.12, −0.10; *p* < 0.0000; cumulative: β = −0.15 ± 0.04, 95% CI: 0.00, 0.72; *p* < 0.0000). Strong statistically significant correlations were also found between ANS recovery metrics and maximum running speed. Specifically, baseline HR was negatively associated with maximum speed (β = −0.45 ± 0.12, 95% CI: −0.70, −0.19; *p* = 0.001). Both metrics of HRV, rMSSD and SDNN, were significantly and positively associated with maximum running speed: (β = 0.32 ± 0.09, 95% CI: 0.14, 0.50; *p* < 0.03 and β = 0.35 ± 0.11, 95% CI: 0.13, 0.57; *p* < 0.02).

## 4. Discussion

The purpose of this study was to examine the association between ECL endured across 13 weeks of in-season training and ANS recovery and function among D1 football players. We hypothesized that higher ECLs endured across weekly in-season training would elicit a negative influence on maximum running speed. The major findings of this study were that among “starters”, (1) both acute and cumulative exposures to high exercise cardiac loads were negatively associated with maximum running speed achieved during in-season training, (2) deteriorating maximum running speed was strongly associated with higher baseline HRs, and (3) HRV (rMSSD and SDNN) were positively associated with maximum running speed.

One novel aspect of this study was the strong, negative associations observed between both acute and cumulative exposures to high ECLs and maximum running speed during 13 weeks of in-season training. Specifically, for athletes in “heavy-running” positions (e.g., wide receiver and tight end), acute and cumulative exposures to high ECLs during in-season training negatively impacted their performance with linear, progressive reductions in maximum running speed. In collegiate football, short sprints at near maximal or maximal speed are critical to a team’s offense and defense, significantly influencing their overall game performance and outcome. Similar to the current study, several studies previously documented decrements in sports performance consequent to excessive acute and chronic exercise training loads. However, most were reported among adult rugby and soccer players in European countries. These sports represent a small fraction of collegiate athletes in the United States as opposed to collegiate football, which accounts for 25% of all NCAA athletes [16]. Moreover, the outcomes of these studies were more focused on soft-tissue injuries, training, and game absenteeism, and less so on performance-based outcomes (e.g., speed, power) [18]. The negative impact on performance-based outcomes is critical to detect as it likely precedes an injury that significantly disrupts physical movement (e.g., muscle strain/tear, ligament strain), requiring passive recovery and rehabilitation [24]. With that, the strong relationship between ECL and maximum running speed observed in this study highlights the potential utility of ECL as a monitoring tool for optimizing performance. ECL quantifies the physiological tolerance of each athlete, using the total exercise load endured by the cardiac muscle during training (HR x duration). Using that physiological feedback provides coaches with a non-invasive measure to program more effective pre- and in-season training regimens and possibly prevent significant decrements in sports performance, non-functional overreaching, and overtraining.

Interestingly, the current study also demonstrated a strong negative association between next-day baseline HR and maximum running speed. That is, athletes exhibiting higher next-day baseline HRs, on average, showed larger decrements in their maximum running speed. This observation possibly suggests that deteriorating speed may manifest from insufficient recovery of the ANS. Given the paralleling negative association between ECL and maximum running speed reported in the current study, the suboptimal recovery of the ANS could be consequent to acute and cumulative exposures to high ECLs. Established evidence demonstrates the negative neurophysiological consequences of high training loads and insufficient recovery. For example, studies show excessive training partially impairs neural signaling (e.g., firing rate) [12], mitochondrial function [13], glucose tolerance [10], skeletal muscle repair [11], etc., all of which require at least 48–72 h or more for full recovery [9]. Without sufficient recovery, ANS function may begin deteriorating and negatively affect the contractile properties of skeletal muscle (e.g., shortening velocity) and performance outcomes such as maximum running speed. Monitoring the relationships between ECL, ANS recovery/function, and performance is likely most critical during the in-season as in addition to exercise training, athletes perform supraphysiological efforts during 3- to 4-h-long competitions. The reductions in maximum running speed observed in this study are the antithetical outcome to the primary goal of in-season training, which is achieving peak athletic performance. For this purpose, sports performance experts recommend substantially decreasing volume and intensity variation in weekly training to focus on maintaining the athletes’ level of fitness and refining sport-specific movement patterns [6,25]. The decrements in performance in this study may highlight the suboptimal translation of recommendations to the real-world sports realm.

Analogous to our baseline HR finding, rMSSD and SDNN were positively associated with maximum running speed. That is, on average, athletes running at lower maximum speeds likely exhibited lower HRV, an indication of insufficient ANS recovery 24 h post-training. Several studies previously documented lower HRV, including rMSSD and SDNN, immediate and short-term post-intense exercise (0–12 h). However, HRV typically returned to baseline values within 24 h [26,27,28]. The discrepancies between the current study and others are potentially attributed to different types of sports (e.g., running and cycling vs. strength and power), duration and frequency of high-intensity training, and sufficient rest intervals between and within sessions. The lower HRV observed *in tandem* with lower maximum running speeds observed in this study aligns with the nature of the off-, pre-, and in-season training programs and the negative association found between baseline HR and maximum running speed. Football practices, across all seasons, were typically long in duration (~2 to 4 h), occurred 5 to 6 times per week, and included several high-intensity training sessions. As such, the nature of these sessions, specifically during the in-season, likely explains our HRV findings. Additionally, the negative association between next-day baseline HR and maximum running speed further confirms this observation. In healthy populations, baseline HR is inversely associated with HRV such that higher baseline HRs correlate with lower HRV. In the current study, this relationship is also observed [29]. For example, in Figure 2 and Figure 3, at the same maximum running speed (e.g., 12.0 mph), higher baseline HRs and lower HRV values are observed, both indicating some degree of ANS dysfunction.

It should be acknowledged, however, that although the metrics referenced in this paper (speed, baseline HR, etc.) correlate, they should not be viewed as surrogates for one another. For instance, while the findings for both metrics of ANS recovery (RMSSD, SDNN, baseline HR) were consistent, each metric provides different physiological information [30]. This is important as athletes exhibit high inter-individual variability in their physiological response and tolerance to training [31]. As such, it is strongly recommended that coaches utilize a holistic approach to monitoring and evaluating their athletes in an effort to prevent deterioration and maximize performance.

### 4.1. Strengths and Limitations

There are several strengths of the current study. First, this study employed a prospective design in a natural setting, which allowed for stronger evidential conclusions and unique insight into collegiate football training and its potential consequences on athletic performance. Second, the ECL metric more accurately measured the total physiological (internal) workload endured by athletes during training as opposed to other methods quantifying workload using sets, repetitions resistance loads, etc. As such, ECL may be a more effective tool for monitoring athletic performance and preventing ANS deterioration. Third, maximum running speed was measured objectively using a device measuring inertial movement. This likely provided a more accurate measure of maximum speed compared to other metrics such as field-based testing and global positioning systems. This study also has a few limitations. First, our sample only included collegiate football players from one D1 university in a single geographical location, limiting the generalizability of our findings. Second, maximum running speed, ECL, baseline HR, and HRV were not collected during regulation games which did not allow us to fully quantify the physiological load endured by the athletes during a given in-season training week. As such, it is unclear as to whether the physiological load of the game influenced the ECL of the subsequent week’s training sessions and vice versa. However, the game-day physiological load was indirectly measured as the data were collected during “in-season” training. Third, the small sample size precluded our ability to analyze the correlations between ECL, HR recovery, and maximum speed by football position. However, the inter-position variability in movement patterns was reduced as only positions with “heavy running” were included in the analyses. Lastly, other factors potentially affecting ANS activity were not accounted for such as sleep, ergogenic aids, and psychological stress.

### 4.2. Practical Implications

This study highlights the importance of coaches appropriately designing exercise training programs during the in-season. Because the in-season includes many regulation games, athletes endure significantly greater physiological loads nearly every week. As such, strength, and conditioning coaches, as recommended by sports performance experts, must dramatically modify their training programs to provide sufficient rest following games, yet simultaneously provide sufficient physiological stimulus to maintain fitness levels. By optimizing training programs throughout the year, sports performance outcomes, such as maximum running speed, can be improved.

## 5. Conclusions

In conclusion, the current study demonstrated that exposure to both high acute, and more importantly, cumulative ECLs, may negatively influence sports performance, specifically maximum running speed. Additionally, the observed decrement in running speed may be a manifestation of a deteriorating ANS, an early warning sign of overtraining. As such, it is imperative that coaches account for the increased physiological load of games thus, optimizing in-season training programs that improve sports performance. Further research is needed to continue identifying optimal training: recovery ratios during off-, pre-, and in-season phases for many sports. We recommend that future investigations monitor, year-round, the physiological loads of training programs to inform coaches of the best practices for preparing for off-, pre-, and in-season training, and providing adequate recovery. Importantly, researchers should also identify ineffective sports training programs that lead to declines in performance.

## Figures and Tables

**Figure 1 jfmk-08-00134-f001:**
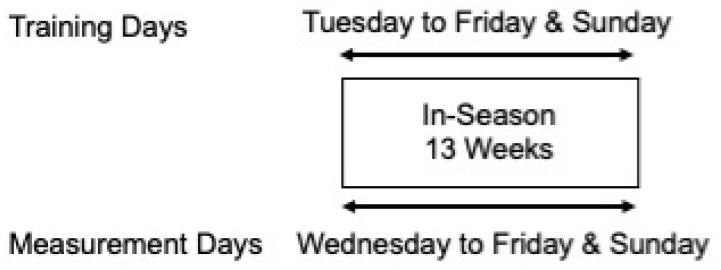
Schematic of 13-week In-Season Collegiate Football Training Program.

**Figure 2 jfmk-08-00134-f002:**
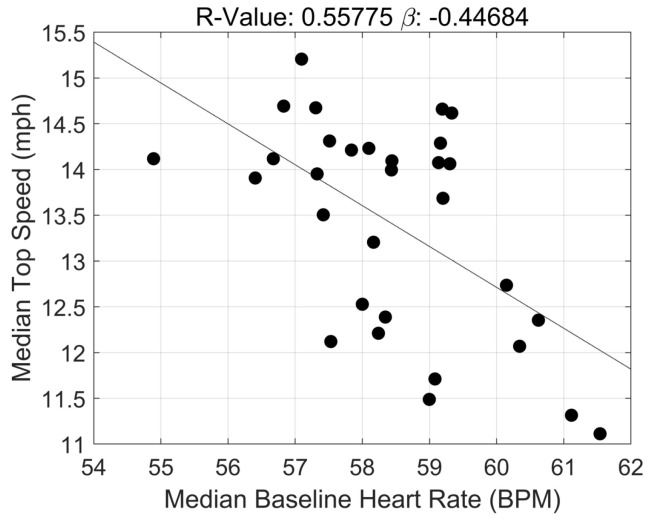
Correlation Between Next-Day Baseline HR and Maximum Running Speed in D1 Football Athletes.

**Figure 3 jfmk-08-00134-f003:**
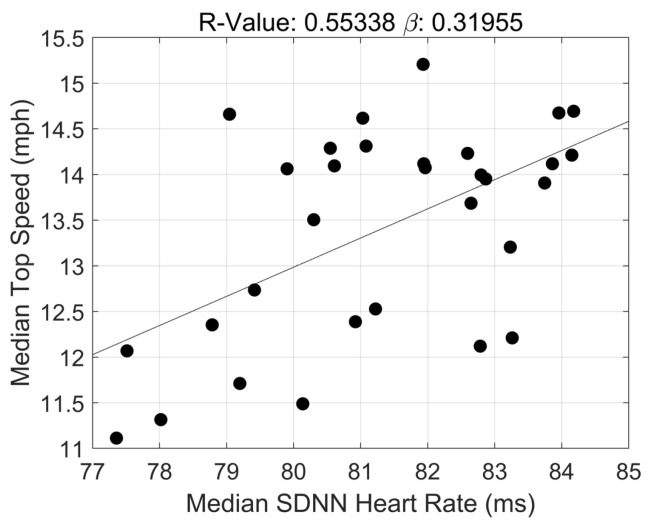
Correlation Between HRV (SDNN) and Maximum Running Speed in D1 Football Athletes.

**Table 1 jfmk-08-00134-t001:** Descriptive Characteristics of Starting Football Athletes (n = 30).

	Mean (SD)	Median (Min, Max)
Age (years)	20.2 (1.5)	20.0 (18.0, 23.0)
Anthropometrics		
Weight (kg)	94.38 (9.7)	92.97 (77.1, 112.5)
Height (m)	1.85 (0.06)	1.84 (1.75, 1.86)
Body Mass Index (kg/m^2^)	27.6 (2.3)	27.5 (23.7, 32.5)
Race/Ethnicity (%)		
NH White	6.7	
NH Black	80.0	
Other	0.0	
Hispanic	13.3	
Football Position (%)		
Cornerback	16.7	
Defensive Back	3.3	
Linebacker	16.7	
Running Back	23.3	
Safety	6.7	
Tight End	13.3	
Wide Receiver	20.0	

NH = non-Hispanic; SD = standard deviation; min = minimum; max = maximum; kg = kilogram; m = meter.

**Table 2 jfmk-08-00134-t002:** Average Training Load, ANS Recovery and ANS Function Among Starting Football Athletes.

	Mean (SD)	Median (Min, Max)
Exercise Cardiac Load *		
Daily (acute exposure)	21.8 (4.6)	23.7 (8.4, 34.8)
Weekly (cumulative exposure)	108.7 (22.8)	114.9 (43.9, 159.8)
ANS Recovery		
Baseline HR (bpm)	60.9 (8.6)	59.8 (48.8, 112.2)
SDNN (bpm)	81.3 (2.0)	81.2 (77.4, 84.2)
rMSSD (bpm)	70.1 (1.7)	70.1 (66.2, 73.4)
ANS Function		
Maximum Running Speed (mph)	17.3 (1.4)	17.2 (15.0, 22.0)

Expressed in the total number of heartbeats; bpm = beats per minute. * average training HR⋅session duration.

**Table 3 jfmk-08-00134-t003:** Adjusted Regression Coefficients for the Association Between ECL and ANS Recovery and Function (Maximum Running Speed [mph]).

	β (SE)	95% CI	Adjusted R^2^	*p*-Value
Exercise Cardiac Load *				
Daily (acute exposure)	−0.11 (0.00)	[−0.12, −0.10]	0.64	0.0000
Weekly (cumulative exposure)	−0.15 (0.04)	[−0.00, 0.72]	0.73	0.0000
ANS Recovery				
Baseline HR (bpm)	−0.45 (0.12)	[−0.70, −0.19]	0.56	0.0011
SDNN (ms)	0.32 (0.09)	[0.14, 0.50]	0.55	0.0287
rMSSD (ms)	0.35 (0.11)	[0.13, 0.57]	0.52	0.0151

CI = confidence intervals; bpm = beats per min; ms = milliseconds; mph = miles per hour. * average training HR⋅session duration.

## Data Availability

Data presented in the current paper are available upon request.

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
