# Peer review of "Exercise Cardiac Load and Autonomic Nervous System Recovery during In-Season Training: The Impact on Speed Deterioration in American Football Athletes"

_jfmk, 2023, doi:10.3390/jfmk8030134_

Round 1
Reviewer 1 Report
The revised article analyzes "Exercise cardiac Load and Autonomic Nervous System Recovery" being a very interesting topic for in competitive sport and high performance that demands maximum speed of displacement over time.
The abstract correctly describes the parts of the work, the introduction is good and adequate and correctly links with the objective and methodology of the research. The methodology and results presented are supported in a well-designed research process.
The discussion about the results is very successful in comparison to previous studies and situations, accepting on the one hand the author the limitation of the sample and not being able to contemplate in the results the contributions of the competition.
From the point of view of this reviewer I highlight the strengths of the study since it was carried out without alteration of the sports process, with non-invasive methods, with a high quality sample and difficult access to researchers, in addition the temporality in the development of research covering a full season provides a high value also.
The bibliography is adequate and presented and distributed in the correct article.
However, the authors are given two indications in order to improve this article.
Line 29 acronym ECL has not previously been defined if it is subsequently defined in line 111.
In line 74, the statement "this remain unclear" should perhaps be accompanied by a reference to that effect.
In the protocol of the measurement of "maximum running speed" in line 139 describes a process for obtaining appropriate data that would be desired by the reviewer to know if there is any previous reference or study when applying that protocol.
Accepted with small changes.
Reviewer 2 Report
This study aimed to examine the association between exercise cardiac load performed during 13 weeks of in-season training and the autonomic nervous system recovery and function in among college American football players.
The manuscript is very cler, very weell written and this reviewer doesn't have further requirements to ask the authors
Reviewer 3 Report
General comments
The study is simple and does not involve complex measurements. The rationale is partially adequate. Some references are missing which need to be incorporated. It is necessary to present the "content" of the training together with the internal load. Were all athletes submitted to the same training? Did everyone participate in training where speed was a primary factor? In this sense, it is necessary to make it clear in the study why no test to assess speed was adopted. The metric used is weak. The environment was not controlled for speed assessment, which, as previously mentioned, may be dependent on the proposed training. The results can be further explored. I could not easily find the sample n. Therefore, I recommend performing the correlations by positions, considering the specificities of the training. The figures need to be redone in environments suitable for this purpose.
Specific comments
Lines 42-43 - It is not clear what ANS recovery refers to. First, it is necessary to explain the reasons for your exhaustion/fatigue/overcoat/use, and so on.
Line 46 – Do not use “etc”.
Line 47 – Change “explosive” to “power”.
Line 49-51 – Again, it is not clear what ANS recovery refers to.
Line 56 – Some citation is required.
Line 56-57 – Same here.
Line 57 – “Sport performance experts” Change to “Researchers”, or something similar.
Line 61-63 – Rewrite this sentence to improve clarity.
Line 73 – Some citation is required.
Line 84 – Create a high-quality figure explaining the experimental design. Insert the collection points, as well as the measured parameters.
Line 85-86 - Please provide more details. At which stage of the competition were the collections carried out? How long did this championship last? Explain why the 3-week clipping. Is this the total period of the competition?
Line 87-88 - …”cardiac function metric: exercise cardiac load.” The meaning was lot here.
Line 88-89 “ANS recovery was measured using baseline heart rate (HR) and heart rate variability (HRV)” Provide some sentences in the introduction explaining the gold standard variable for the ANS recovery. Further, explain here why these variables were chosen.
Line 89-90 – “…and ANS function was estimated via the athletes’ top speed reached during weekly trainings.” Provide the scientific basis (several studies published in journals with a high impact factor) that attributes the reduction in the "reached speed" to ANS function.
Line 92 – How many?
Line 95-96 - During the thirteen weeks, did the authors confirm whether the nominated athletes were in fact starters? Not enough, explain why the other athletes were not included. Considering the rationale for the introduction, reserves should not present a reduction in developed speed, and consequently a greater reduction in ANS. I particularly disagree with this point, considering that coaches/physiologists must adjust the training load individually, preventing reserves from going into detraining.
Line 114-115 – It is imperative to provide strong references.
Round 2
Reviewer 3 Report
---While most of my questions were answered, some were not. It is imperative to make all modifications.
-Provide sample size in line 97, not in line 172.
-In limitations, provide explain why correlation by position were not performed.
-Remove the term explosive. Muscles do not explode.
-Provide a high-quality figure explaining the experimental design. Also, improve the quality of all figures.
Round 3
Reviewer 3 Report
The authors answered all my questions.